# From EHR to PHR: let's get the record straight

Joshua D Symons,[1] Hutan Ashrafian, [1] Rachel Dunscombe,[2] Ara Darzi[1]

[1]Institute of Global Health Innovation, Imperial College London, London, UK
[2]Salford Royal NHS Foundation Trust, Salford, UK

**Correspondence to**
Mr Joshua D Symons;
j.symons@imperial.ac.uk

## ABSTRACT

This article reflects on the changing nature of health information access and the transition of focus from electronic health records (EHRs) to personal health records (PHRs) along with the challenges and need for alignment of national initiatives for EHR and PHR in the National Health Service (NHS) of the UK. The importance of implementing integrated EHRs as a route to enhance the quality of health delivery has been increasingly understood. EHRs, however, carry several limitations that include major fragmentation through multiple providers and protocols throughout the NHS. Questions over ownership and control of data further complicate the potential for fully utilising records. Analysing the previous initiatives and the current landscape, we identify that adopting a patient health record system can empower patients and allow better harmonisation of clinical data at a national level. We propose regional PHR 'hubs' to provide a universal interface that integrates digital health data at a regional level with further integration at a national level. We propose that these PHR hubs will reduce the complexity of connections, decrease governance challenges and interoperability issues while also providing a safe platform for high-quality scalable and sustainable digital solutions, including artificial intelligence across the UK NHS, serving as an exemplar for other countries which wish to realise the full value of healthcare records.

## BACKGROUND

The importance of implementing integrated electronic health record (EHR), increasingly referred to as electronic patient records (EPRs), as a route to enhance the quality of health delivery in the UK National Health Service (NHS) were outlined as early as 1998.[1] The benefits of universal access for patients, clinicians, researchers, policymakers and administrators include (1) increasing confidence and convenience through streamlined access to information, (2) improving clinical outcomes through enhanced care communication, (3) better science and decision making through access to real-time evidence and (4) enabling planning and productivity gains through increased efficiency.[1] In 2014, it was proposed that by March of 2018, all individuals should have both viewing and editing capabilities for their own health record.[2] This proposal aligns with the definition of a personal health record (PHR) by the Royal College of Physicians as 'digital tools that allow a citizen to interact with health and social care services, have access to the clinical content in their record, capture, record and if they wish, share their own data with clinicians and others'.[3] PHRs therefore provide an integral part of health record integration that is not currently provided by EHR systems alone. Furthermore, PHRs have been shown to contribute to health awareness and could be key in empowering patients to take direct control of their own healthcare.[4] However, despite the widespread availability of multiple PHR solutions in the UK, 2018 has now passed without substantial adoption of PHRs and empowerment of patients through meaningful interaction with their own healthcare data.

Although there has been growth in the number of PHR providers in almost all regions of the UK, PHR uptake by patients and organisations is still remarkably slow.[5] In response to a survey in 2016, less than 500 000 people were individually registered users of PHR while 1377 organisations, including 122 NHS Trusts, 16 Clinical Commissioning Groups (CCGs), 1184 general practice surgeries (GPs), 33+ local authorities, 15+ social care providers and 7 health boards were listed as organisational users of PHR.[6] However, it is not clear if single organisations have integrated with multiple PHR providers thereby conflating the totals and making them appear larger than they are, for example, counts reflecting that nearly half of all NHS Trusts are organisational users.[6] Slow uptake has been attributed to barriers in (1) sharing of data at a local level, (2) clinical aversion and reluctance, (3) patient awareness and (4) technical integration with local information systems.[6] Despite improvements in data sharing mandates, namely through the Caldicott principles, the first challenge remains heavily influenced by the second and third challenges as sharing of data is inhibited where there is a lack of clinician and patient endorsement.[6] Furthermore, PHR solutions

**Table 1** Examples of different care providers, PHR and EHR providers and exchange protocols used within the UK.

| Type | Count |
|---|---|
| Care organisations | 1. (207) CCGs<br>2. (152) Acute specialist and non-specialists trusts<br>3. (54) Mental health trusts<br>4. (35) Community providers<br>5. (10) Ambulance trusts<br>6. (853) For-profit and not-for-profit independent sector organisations<br>(7148) GPs |
| PHR providers | (29) Independent providers recorded by RCP |
| GP EHR providers | (4) EMIS, TPP, INPS and Microtest |
| Trust EHR providers | (7) Allscripts, Cerner, Epic, Intersystems, Lorenzo, Meditech and OpenEHR |
| Exchange protocols | (6) HL7, FHIR, IHE, CDA, XDS and DICOM |

The list is non-exhaustive.
CCG, Clinical Commissioning Group; EHR, electronic health record; GP, general practice surgeries; PHR, personal health record.

have the ability to address the first three challenges through patient control of records and bringing value for clinicians and patients yet this ability is hampered by the formidable challenge of local integration. Given that the majority of existing UK PHR solutions are reliant on direct integration with existing EHR solutions, local information exchange remains a key inhibitor to gaining value from PHRs.[6]

Despite a large amount of research in EHR interoperability, there is a paucity of evidence regarding barriers for EHR integration with PHR, that is, PHR interoperability. The prospect of universally connecting PHR systems is daunting considering the sheer number of individual CCGs, NHS Trusts, GPs, PHR and EHR vendors and exchange protocols (table 1 and figure 1). Established node connection formulae demonstrate that 8500 organisations could require over 36 million individual connections to provide real-time connectivity of records for patients at any location.[7 8] This does not take into account additional organisations such as laboratories, social and community care. Further addition of PHR providers, personal devices and applications will exponentially increase this complexity. Solutions have been proposed to deal with some of this complexity as part of national programmes but these fall short of implementing the benefits of PHR. Therefore, both technical and socio-technical barriers for information exchange between health record systems continue to present the biggest barrier for enabling these benefits.

## REDUCING BARRIERS FOR INFORMATION EXCHANGE

Local Health and Care Record Exemplars (LHCREs) have been created in the NHS with the same proposed benefits outlined in 1998 through integration of care records, such as primary care with secondary, mental health and social at a regional level.[9] Although LHCREs aim to achieve real-time data sharing of both identifiable and de-personalised data for direct care, the scope of these integrated records does not universally include provision of linked data for research or patient access to read and write to integrated records.[9] Approaches to governance of data in LHCRE are not universal, with some initiatives putting control of research and sharing permission with GPs and Academic Health Science Networks while others form independent research advisory groups. Some LHCREs

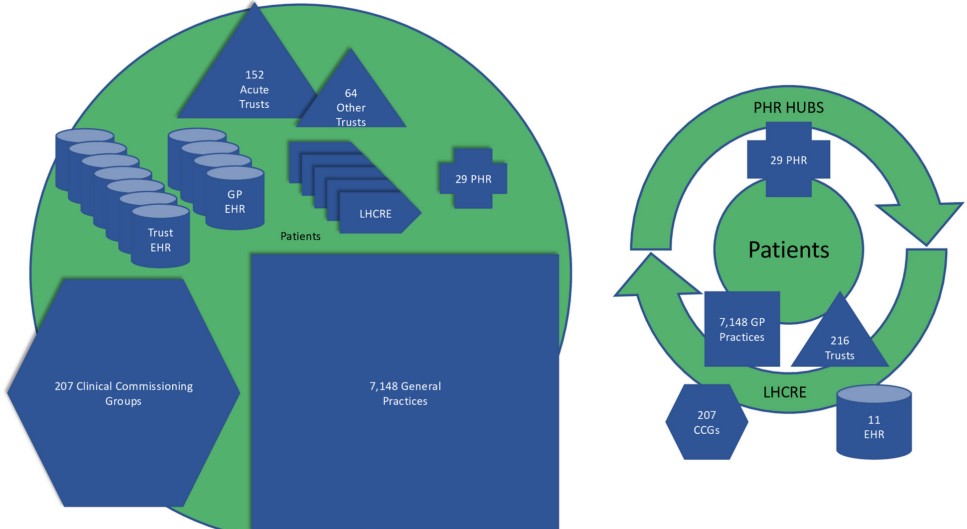

**Figure 1** PHR hubs would transform the complex ecosystem (left) including high patient and provider burden to a simpler ecosystem (right) centred around the patient increasing choice and system flexibility. CCG, ClinicalCommissioning Group; EHR, electronichealth record; GP, generalpractice surgeries; PHR, personal health record.

are implementing PHR repositories at a regional level but this is not universal and different LHCREs are embracing different technologies.[10] This creates regional variation in data sharing practices, record access and approved uses of data. Furthermore, resources are limited, with some LHCRE funding only covering £7.5 million of the estimated £90 million 5-year cost, making non-critical features difficult to justify.[10] This raises important questions around sustainability of LHRCEs which have yet to be answered. Therefore, although LHCRE may provide integration of records to improve direct care, patients will not necessarily benefit from access to these records, researchers may not be able to use these linked records to produce evidence and industry may not have access to the benefits of real-time data access to develop new digital interventions. Digital innovation hubs, funded through Health Data Research UK and aiming to increase research using local integrated records, will go some way in addressing use of LHCRE data for innovative research but will not provide patient access or data input into these records.[11]

At a national level, the National Record Locator Service (NRLS), not to be confused with the same abbreviation used for the National Reporting and Learning System, has been proposed to achieve harmonisation of individual health records across the UK.[12] This will be achieved by connecting NRLS to various electronic patient master index (EPMI) systems at care providers which will keep care records up to date. Although this important service will aim to provide a pointer to a single source of truth record for each patient, it will not guarantee the quality of that record or provide integration of records for non-clinical sources of patient data. The proof of concept for the service will also not include retrieval of the actual patient records and early features are aimed at providing access for direct care, therefore this service will not be able to provide real-time access to patient records at a national level.[12] NRLS in its current iteration will require that care provider services will be always available for querying using a pull method, but services may not be available resulting in limited access to required records. Connectivity to records at a national level will be dependent on the quality of local EPMI and EHR systems. Such connectivity issues could result in patient safety incidents due to missing records. Furthermore, there is a need to consider other sources of data relevant to personal health such as income, education and other data which patients may be able to share easily given the right mechanism. This has given rise in 'smart cities' to the concept of a person master index (PMI) as an alternative to an EPMI.

The newly released *NHS App* will allow patients to check symptoms, book GP appointments, manage repeat prescriptions, view their own primary care record and manage data sharing preferences.[13] These features are enabled through a 'broker' system which translates individual health record provider implementations.[14] Although this mobile application will create an excellent baseline entry point for every NHS patient to view their GP records, it does not include any ability to edit or even view the entire patient record. Other features will allow patients to interact with portions of the health and social care system but listed benefits indicate that this tool will provide a passive consumption service for commissioned digital services, without allowing patients to integrate digital services they have already chosen. Despite improving access to NHS services and providing a starting point for universal primary care record access, the NHS application in its current form will not provide a full universal PHR solution in its initial form and will overlap with functionality provided by existing EHR portals and PHR platforms.[15–17] Although the NHS App cannot replicate existing PHR solutions and may not be designed to replace them it is unclear at this time how the release of this application might impact the ability of PHR providers to market their products and services. It is also unclear on how patient opt-out for data sharing in the NHS App will impact data sharing at a regional level as patients might believe that using the application to opt-out of data sharing will stop any sharing of their data in any system, resulting in a lack of clarity and transparency for patients and organisations alike with potential loss of trust across all stakeholders.

PHR which enables full two-way integration of both care provider health record data and patient generated data with patient control of data sharing, is the only current proposed health record solution which provides a choice to patients for inputting into and controlling their own health records. PHRs also provide benefit by providing digital intervention data generated by patients to clinicians as well as providing a missing piece of research data necessary to evaluate the true patient journey. Current PHR provider models are mainly centered on integrating with local EHR systems at a care provider level, which has been shown to be more beneficial than standalone PHR systems, although both models have been reported as offering healthcare benefits.[6] Although this meets the criteria of providing patients access, it does not provide patients a choice in features of different PHR providers or join data across multiple providers. Integration is also done using bespoke interoperability with local providers one solution at a time. To provide choice in PHR features, integration with multiple PHR providers is required by single institutions. This increases the burden and cost to these institutions to provide multiple PHR solutions and makes universal adoption of PHR more difficult as patients and clinicians are unable to determine what benefit different PHR solutions may provide. Stand-alone PHR systems which enable patient input will not necessarily be integrated with PMI systems and therefore data from these systems will not be available to NRLS, making the patient responsible for communicating relevant information to clinicians and exacerbating the existing burden for patients.

## LEARNING FROM THE PAST
The dangers of trying to individually connect EHRs nationally and regionally while simultaneously increasing

adoption of PHR with the current system structure are substantial. This could lead to record fragmentation which is an established concern and possibly lock data into another silo, unavailable to the persons or systems which need it.[18] Additionally, there could be an increasing disparity between the information patients think is available to providers compared with what exists at the point of care. NRLS would need to connect to more systems with less certainty over the range of standards which will be required to locate records. Although the PHR adoption toolkit provides guidance for most standards, implementation of these interoperability standards will inevitably vary.[19] Currently, there is also no solution for universal integration of personal device data into most EHR systems. Some PHR systems provide this functionality, but monitoring of digital interventions such as prescribed mobile applications will not be possible where personal device data is not being integrated back into clinical records. As suggested in a 2018 REFORM report on data value, integration of such data should be explicitly included in NHS procurement contracts to ensure that digital interventions can be tested for safety and efficacy.[20] Furthermore, the benefits of integrating patient generated data into clinical records have been clearly outlined in the Topol review.[21] Universal integration of this form of patient data into clinical records will be a large challenge for any EHR provider.

## A POTENTIAL SOLUTION

To overcome the barriers of universal record integration and PHR adoption, we propose a new model of connecting systems with a PHR 'hub' providing integration at regional levels for a single 'source of truth' PHR (figure 1). The PHR hub could operate in a similar fashion to the 'broker' system used by the *NHS App* to communicate with different implementations of GP systems, translating data across EHR and PHR providers thereby reducing complexity immensely while simultaneously moving towards universal choice and control for patients. Complexity, barriers to interoperability and the potential for vendor lock-in would be further reduced by requiring open standards in the PHR hub.[22] Such an initiative would naturally complement existing LHCREs, especially if surfaced or directly integrated at a regional level, reducing variability for LHCRE implementation and enabling LHCREs to focus on critical record integration. Combining this with patient controlled PHRs would decrease governance burden for organisations as patients themselves would control records or be able to delegate this to care providers as needed. Furthermore, each regional PHR could facilitate universal patient validated PMI by bridging local PMI systems to keep patient details up to date. This would then ease requirements for the NRLS.[12] PHR hubs could also separate care record service from care settings allowing for better security and easier data migrations. PHR hubs would provide a single integration point between multiple PHR and EHR providers taking away the burden of integration from the care provider. Finally, universal PHR accessibility would allow PHR vendors to market directly to patients supported by clinician conversations around the benefits, possibly providing a far more efficient method of increasing patient adoption and providing an easier path to self-care.

## CONCLUSION

The ambition of achieving the potential of PHRs has long been established. There have been major achievements towards understanding the value of increased information flow, especially for patients, based on recommendations of the Wachter report.[23] This includes the appointment of Clinical Chief Information Officers across the NHS, the launch of the NHS App and new training initiatives such as the NHS Digital Academy.[24] Despite these achievements, barriers such as a lack of EHR systems, lack of PHR functionality in existing EHR systems, the controversial legacy of top-down implementations such as the costly UK National Programme for Information Technology and continuous interoperability issues have presented formidable opponents to this ambition.[24] Enabling true patient empowerment requires that organisations are able to universally connect to PHR providers and that patients are able to select the PHR provider they prefer. Understanding and addressing facilitators and barriers of PHR adoption along with barriers for PHR interoperability should be a key focus for the NHS and researchers. Only where PHR adoption is driven by patients will PHR access fully satisfy the requirements of its intended users. This requires facilitating a market place for PHR providers who can address patient demand using their own data. We have proposed a new call for regional PHR hubs to provide a universal PHR interface and integrate EHR systems using the existing broker system already provided by the NHS App as an example.[10] As suggested in the Watcher report, we propose that such hubs should be implemented in regional settings to encourage local ownership.[23] This will reduce the complexity of connections, decrease governance challenges and interoperability issues while also providing a safe platform for high-quality scalable and sustainable digital services, such as personal artificial intelligence, across the NHS. Achieving standardised universal PHR adoption with EHR data in an integrated record in the NHS will unlock the true value of the UK's integrated healthcare records and can serve as an example to other countries which wish to unlock the true potential of their healthcare records.

**Contributors** JDS: conception, design, final draft, approval and agreement to be accountable. HA: conception, design, final draft, approval and agreement to be accountable. RD: design, final draft, approval and agreement to be accountable. AD: design, final draft, approval and agreement to be accountable.

**Funding** Infrastructure support for this research was provided by the NIHR Imperial Biomedical Research Centre (BRC).

**Competing interests** None declared.

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
