## [Reviewer comments · BMJ Open]

ARTICLE DETAILS

TITLE (PROVISIONAL)	From EHR to PHR: Let's Get the UK Record Straight
AUTHORS	Symons, Joshua D; Ashrafian, Hutan; Dunscombe, Rachel; Darzi, Ara

VERSION 1 - REVIEW

REVIEWER	Will Dixon The University of Manchester, UK
REVIEW RETURNED	13-Mar-2019

GENERAL COMMENTS	The article provides a useful overview of the opportunities and challenges of PHRs in the UK, with a proposed solution to overcome some of the challenges. One key thing that I think would help is signposting at the start of the article to more explicitly link the various sections together, giving a clear overview of the article and the list of problems that will be discussed. Despite being relatively short, subtitles might help, labeled to align with the key message of each paragraph. I hope the following specific comments will be useful in helping to strengthen the article. • Line 31. "Slow uptake has been attributed to (i) ..." Each of these reasons could usefully be expanded. Governance approval, for example, is too generic for the general reader to grasp what the particular issues are. Whilst detailed expansion might upset the flow of the article, listing one or two of the highest barriers per topic would be useful.• Line 33. "Challenges in governance are being overcome...". Again, an example of how this is enabling the barrier to be overcome would help• Paragraph 3 has an important focus on the use of EHR data for research. The preceding two paragraphs did not explicitly mention research e.g. as an important stakeholder in line 6, or in the list of benefits in the subsequent sentence (accepting that real-time evidence may be referring to research rather than the provision of patient data to improve decision making). As such, I found the paragraph going down a line of discussion I hadn't expected. This might be helped by some signposting earlier in the article to explain this is coming and how it fits, or some subheadings so the reader knows we are moving onto discussion about research.• The same argument applies to subsequent paragraphs. Each topic is important, but I was needing to piece them together as I read. An introduction to these important topics at the end of the
--

	first paragraph, and a description of how they fit together, would help.  • The challenges of implementing patient opt-out of data sharing within the NHS App are an important point. Differentiating data sharing preferences for clinical care and research is something else that might be mentioned, as are the consequences of “the lack of clarity and transparency”, namely a loss of public trust in this sensitive area. • Line 51. The integration of patient-generated health data/ personal device data is a very important challenge, and a major opportunity for PHRs. I would welcome this section being expanded to make clear what the potential benefits are, but also the difficulties in making this happen. This is usefully summarized in other articles such as the ONC report (https://www.healthit.gov/sites/default/files/pghd_brief_final122013.pdf) • “The iterative evaluation of digital interventions” is a little difficult to understand without further explanation. • Please provide a reference for the REFORM report (line 53) • The concluding paragraphs provide useful guidance for how many of the challenges might be overcome. I did wonder whether the authors might be explicit about what are the top priorities or requirements for implementing such as system – what needs to be done to make this happen eg user authentication, data sharing preferences to include clinical care and research etc? What are the current ‘known unknowns’, informing a research agenda in this area.
--	---

REVIEWER	Brian Dixon Regenstrief Institute, Center for Biomedical Informatics, Indianapolis, Indiana, USA
REVIEW RETURNED	15-Mar-2019

GENERAL COMMENTS	The abstract promises a clear analysis of the health IT landscape in the UK followed by a plan for PHR access to enable health system transformation. Unfortunately this commentary does not deliver on the promise of the abstract. The paper asserts many generalities about EHR systems and other HIT components in the UK. Then it only briefly introduces the concept of a PHR 'hub' before concluding. This leaves the reader with more questions than answers. A major challenge with the article is the lack of clear explanation of the concept of a 'PHR hub.' The figure is unhelpful as it simply re-arranges and shrinks the images on the left. It does nothing to explain what a hub is or how it functions. The paragraph that describes the hub provides few details about what the hubs will do besides offer open standards. Would these hubs function differently than interface engines between EHRs and other HIT systems? Furthermore, would these hubs cache data or would they simply exchange data? It is unclear how these will function and how they will address the gaps in the existing infrastructure. Another important challenge with this paper is it assumes the reader has expertise in informatics and interoperability within the context of the UK. Many acronyms and jargon are used to make points about how the current infrastructure is insufficient to achieve the goals of an integrated health system. It would be hard
--

	for patients to get much out of this paper. Even policymakers would find it challenging. A final challenge with the paper is the limited explanations of concepts and points. Much of the article is commentary, and the comments suggest the existing system simply is not working. Yet it is not clear why it isn't working. There are a number of software tools and standards available for use within provider offices, facilities, trusts, and PHR companies. How would these technical components interact with the hubs, and why would they connect to hubs differently than they connect to one another? Given the list of standards in the Table, it is unclear how they are being used now and why we would need new, open standards. Like the health system it is describing, this article is complex. More clarity and a greater focus on the hubs and precisely how they would address the failures of the current system would dramatically improve this commentary.
--	--

REVIEWER	Sarah Pontefract University of Birmingham, UK
REVIEW RETURNED	20-Mar-2019

GENERAL COMMENTS	The authors discuss the personal health record and how this can feasibly be achieved, with integration with other electronic patient records in the NHS. The authors need to make clear that this paper is a UK perspective, which centres on the National Health Service. The authors should refer to the Topol review in the introductory section of the article. Preparing the healthcare workforce to deliver the digital future (2019). Electronic Patient Record (as opposed to health record) tends to be referred to now in the NHS (for example, throughout the Topol review and by NHS Digital). This will need reviewing throughout the article. Page 3 of 10, Line 9: Expand NHS Page 4 of 10, Line 19: the authors refer to "2018 is now passing" – this needs reviewed as has now passed. Page 4 of 10, Line 33: Avoid the use of 'recent' in the article, as it does not make the manuscript timely (refer to the year if necessary). Page 5 of 10, Line 41: NRLS is used here. I would avoid stating the acronym as the majority of readers of the journal will understand this to be the National Reporting and Learning System for the NHS, and thinking ahead, it will confuse the picture in any key word searches. Page 6 of 10, line 3: The NHS App is discussed at length in the Topol review, so this will need reviewing so that the information is up-to-date. Page 6 of 10, line 53: As above, avoid the use of 'recent'. Page 7 of 10: Line 29: Are the authors referring to failures in the NHS here? I find this section a little negative given all the work currently being undertaken by NHS Digital, the introduction of Clinical Information Officers, Digital Academies etc. Again, the
--

	authors need to refer to the Topol review as well. The authors should also reference this section (the “barriers”), as I imagine it is Connecting for Health that is being referred to when “failed top-down” implementation is discussed. The Wachter review (see https://www.gov.uk/government/publications/using-information-technology-to-improve-the-nhs/making-it-work-harnessing-the-power-of-health-information-technology-to-improve-care-in-england , which should also be referenced) makes it very clear that top-down implementation is no longer the approach the NHS is (or will) take to achieve digitisation.
--	---

VERSION 1 – AUTHOR RESPONSE

Reviewer(s)' Comments to Author:

Reviewer: 1

Reviewer Name: Will Dixon

Institution and Country: The University of Manchester, UK

Please state any competing interests or state 'None declared': None declared

Please leave your comments for the authors below

- The article provides a useful overview of the opportunities and challenges of PHRs in the UK, with a proposed solution to overcome some of the challenges. One key thing that I think would help is signposting at the start of the article to more explicitly link the various sections together, giving a clear overview of the article and the list of problems that will be discussed. Despite being relatively short, subtitles might help, labeled to align with the key message of each paragraph.

- o Subtitles have been added to sections as per editors' comments, we have also clarified the argument and overview more clearly in the abstract. We have also expanded some sections enough to be easier for the audience as per suggestions below. It has also been clarified that the article focuses on the challenge of information exchange while touching on other challenges.

I hope the following specific comments will be useful in helping to strengthen the article.

- Line 31. “Slow uptake has been attributed to (i) ...” Each of these reasons could usefully be expanded. Governance approval, for example, is too generic for the general reader to grasp what the

particular issues are. Whilst detailed expansion might upset the flow of the article, listing one or two of the highest barriers per topic would be useful.

- o Clarified challenges and added lines to clarify how first 3 challenges are influenced by the last challenge, which is the primary focus of the article.

- Line 33. “Challenges in governance are being overcome...”. Again, an example of how this is enabling the barrier to be overcome would help

- o Moved this portion to the paragraph above and clarified the interaction of these challenges

- Paragraph 3 has an important focus on the use of EHR data for research. The preceding two paragraphs did not explicitly mention research e.g. as an important stakeholder in line 6, or in the list of benefits in the subsequent sentence (accepting that real-time evidence may be referring to research rather than the provision of patient data to improve decision making). As such, I found the paragraph going down a line of discussion I hadn’t expected. This might be helped by some signposting earlier in the article to explain this is coming and how it fits, or some subheadings so the reader knows we are moving onto discussion about research.

- o Researchers have been added to stakeholders in 2nd line. The paragraph now fits across the stakeholders.

- The same argument applies to subsequent paragraphs. Each topic is important, but I was needing to piece them together as I read. An introduction to these important topics at the end of the first paragraph, and a description of how they fit together, would help.

- o Clarified references to direct care or research or all stakeholders where required.

- The challenges of implementing patient opt-out of data sharing within the NHS App are an important point. Differentiating data sharing preferences for clinical care and research is something else that might be mentioned, as are the consequences of “the lack of clarity and transparency”, namely a loss of public trust in this sensitive area.

- o Clarified as requested

- Line 51. The integration of patient-generated health data/ personal device data is a very important challenge, and a major opportunity for PHRs. I would welcome this section being expanded to make clear what the potential benefits are, but also the difficulties in making this happen. This is usefully summarized in other articles such as the ONC report (https://www.healthit.gov/sites/default/files/pghd_brief_final122013.pdf)

- o Expanded to clarify what we see as the most beneficial implementation of PHR and further associated benefits

- “The iterative evaluation of digital interventions” is a little difficult to understand without further explanation.

- o Have clarified this line to be more lay friendly

- Please provide a reference for the REFORM report (line 53)

- o We have cited this

- The concluding paragraphs provide useful guidance for how many of the challenges might be overcome. I did wonder whether the authors might be explicit about what are the top priorities or requirements for implementing such as system – what needs to be done to make this happen eg user

authentication, data sharing preferences to include clinical care and research etc? What are the current 'known unknowns', informing a research agenda in this area.

o Clarified conclusion to outline key priorities for the NHS and researchers, with clarification in beginning of article that enabling this will help with other challenges for adoption.

Reviewer: 2

Reviewer Name: Brian Dixon

Institution and Country: Regenstrief Institute, Center for Biomedical Informatics, Indianapolis, Indiana, USA

Please state any competing interests or state 'None declared': None declared

Please leave your comments for the authors below The abstract promises a clear analysis of the health IT landscape in the UK followed by a plan for PHR access to enable health system transformation. Unfortunately this commentary does not deliver on the promise of the abstract. The paper asserts many generalities about EHR systems and other HIT components in the UK. Then it only briefly introduces the concept of a PHR 'hub' before concluding. This leaves the reader with more questions than answers.

- A major challenge with the article is the lack of clear explanation of the concept of a 'PHR hub.' The figure is unhelpful as it simply re-arranges and shrinks the images on the left. It does nothing to explain what a hub is or how it functions. The paragraph that describes the hub provides few details about what the hubs will do besides offer open standards. Would these hubs function differently than interface engines between EHRs and other HIT systems? Furthermore, would these hubs cache data or would they simply exchange data? It is unclear how these will function and how they will address the gaps in the existing infrastructure.

o We have clarified that we see this a broker system at least initially with a valid proof of concept to be found in the existing NHS App and its interaction with GP EHR systems.

- Another important challenge with this paper is it assumes the reader has expertise in informatics and interoperability within the context of the UK. Many acronyms and jargon are used to make points about how the current infrastructure is insufficient to achieve the goals of an integrated health system. It would be hard for patients to get much out of this paper. Even policymakers would find it challenging.

o We have clarified that this is UK and NHS centric with clarifications on relevant terms

- A final challenge with the paper is the limited explanations of concepts and points. Much of the article is commentary, and the comments suggest the existing system simply is not working. Yet it is not clear why it isn't working. There are a number of software tools and standards available for use within provider offices, facilities, trusts, and PHR companies. How would these technical components interact with the hubs, and why would they connect to hubs differently than they connect to one another? Given the list of standards in the Table, it is unclear how they are being used now and why we would need new, open standards.

- o We have highlighted this in paragraph 3

- Like the health system it is describing, this article is complex. More clarity and a greater focus on the hubs and precisely how they would address the failures of the current system would dramatically improve this commentary.

- o We have clarified that this article is UK centric and requires an understanding of some of the underlying complexities in NHS information exchange. We have also clarified challenges, existing solutions and what still remains to be done.

Reviewer: 3

Reviewer Name: Sarah Pontefract

Institution and Country: University of Birmingham, UK

Please state any competing interests or state 'None declared': None declared

Please leave your comments for the authors below The authors discuss the personal health record and how this can feasibly be achieved, with integration with other electronic patient records in the NHS.

- The authors need to make clear that this paper is a UK perspective, which centres on the National Health Service.

- o We have clarified this in the article

- The authors should refer to the Topol review in the introductory section of the article. Preparing the healthcare workforce to deliver the digital future (2019).

- o We have referenced the Topol review to support learning from the past.

- Electronic Patient Record (as opposed to health record) tends to be referred to now in the NHS (for example, throughout the Topol review and by NHS Digital). This will need reviewing throughout the article.
- o Have clarified that EPRs are increasingly being used instead of EHR

- Page 3 of 10, Line 9: Expand NHS
- o First abbreviation explained

- Page 4 of 10, Line 19: the authors refer to “2018 is now passing” – this needs reviewed as has now passed.
- o This has been amended

- Page 4 of 10, Line 33: Avoid the use of ‘recent’ in the article, as it does not make the manuscript timely (refer to the year if necessary).
- o This has been amended

- Page 5 of 10, Line 41: NRLS is used here. I would avoid stating the acronym as the majority of readers of the journal will understand this to be the National Reporting and Learning System for the NHS, and thinking ahead, it will confuse the picture in any key word searches.
- o We agree with the reviewers and have clarified that it should not be confused with the existing NRLS with a line in the relevant paragraph.

- Page 6 of 10, line 3: The NHS App is discussed at length in the Topol review, so this will need reviewing so that the information is up-to-date.
- o We have amended to reflect our information which comes directly from implementation level at NHS Digital and NHS England

- Page 6 of 10, line 53: As above, avoid the use of ‘recent’.
- o This has been amended

- Page 7 of 10: Line 29: Are the authors referring to failures in the NHS here? I find this section a little negative given all the work currently being undertaken by NHS Digital, the introduction of Clinical Information Officers, Digital Academies etc. Again, the authors need to refer to the Topol review as well. The authors should also reference this section (the “barriers”), as I imagine it is Connecting for Health that is being referred to when “failed top-down” implementation is discussed. The Wachter review (see <https://www.gov.uk/government/publications/using-information-technology-to-improve-the-nhs/making-it-work-harnessing-the-power-of-health-information-technology-to-improve-care-in-england>, which should also be referenced) makes it very clear that top-down implementation is no longer the approach the NHS is (or will) take to achieve digitisation.
- o We have amended to make more positive and clarify the focus of our article. We have also mentioned the reports and progress the reviewer has kindly pointed out.

VERSION 2 – REVIEW

REVIEWER	Brian E Dixon Richard M. Fairbanks School of Public Health, Indiana University, Indianapolis, Indiana, USA and the Regenstrief Institute Center for Biomedical Informatics, Indianapolis, Indiana, USA
REVIEW RETURNED	28-May-2019

GENERAL COMMENTS	The authors have made considerable progress with the article since the prior version. The comments from the reviewers have largely been addressed with this revision. Overall the commentary calls for the development of "PHR hubs" that aim to integrate EHR and PHR systems to enable interoperability across the complex environment in the NHS. These hubs would provide access to patients and providers for information contributed by providers and patients. While the commentary is much improved, there remain a few minor items to address. 1) While this version of the commentary is clearer, it remains unclear how the LHCREs will interface with the PHR hubs. To this reviewer (someone from outside the UK), the LHCREs aim to integrate data across EHR systems in providers, including GPs. If this is the case, then if the PHR hubs focus on integration of data across PHR systems, then an interface between PHR Hubs and LHCREs would achieve the vision described in the commentary. Yet the LHCREs are not depicted in the Figure. There is some text on the bottom of Page 7 that alludes to this, but it is absent from the architecture. Please clarify and depict the LHCREs in the Figure. Related to this point, it is not clear why the technical standards need to be depicted in the Figure like a GP or PHR. These messaging and technical standards are designed to enable interoperability among information systems; they are not an institution that connects to the nationwide network. Their inclusion in the commentary and Figure seems out of place. 2) The word "carers" on Line 13, Page 7 is odd. Consider replacing with "providers." 3) The abstract mentions artificial intelligence, but this concept is not discussed in the commentary. It also doesn't fit with the main thesis of the essay. Please remove.
---

VERSION 2 – AUTHOR RESPONSE

Below is a point-by-point response:

Feedback below:

1) While this version of the commentary is clearer, it remains unclear how the LHCREs will interface with the PHR hubs. To this reviewer (someone from outside the UK), the LHCREs aim to integrate

data across EHR systems in providers, including GPs. If this is the case, then if the PHR hubs focus on integration of data across PHR systems, then an interface between PHR Hubs and LHCREs would achieve the vision described in the commentary. Yet the LHCREs are not depicted in the Figure. There is some text on the bottom of Page 7 that alludes to this, but it is absent from the architecture. Please clarify and depict the LHCREs in the Figure.

We have clarified the text slightly and the diagram to reflect how LHCREs and PHR hubs could interact though the final relationship might be determined by regional factors for policy and funding.

Related to this point, it is not clear why the technical standards need to be depicted in the Figure like a GP or PHR. These messaging and technical standards are designed to enable interoperability among information systems; they are not an institution that connects to the nationwide network. Their inclusion in the commentary and Figure seems out of place.

We have removed protocols.

2) The word "carers" on Line 13, Page 7 is odd. Consider replacing with "providers."

We have modified this as suggested.

3) The abstract mentions artificial intelligence, but this concept is not discussed in the commentary. It also doesn't fit with the main thesis of the essay. Please remove.

We have modified this to reflect that AI is meant to represent AI in personal digital services, a good example of this is the recently announced ability to use Alexa to access the NHS website.